# Lamin B1 is required for mature neuron-specific gene expression during olfactory sensory neuron differentiation

Crystal M. Gigante[1,2], Michele Dibattista[3,4], Frederick N. Dong[1], Xiaobin Zheng[2], Sibiao Yue[2], Stephen G. Young[5], Johannes Reisert[3], Yixian Zheng[2] & Haiqing Zhao[1]

B-type lamins are major constituents of the nuclear lamina in all metazoan cells, yet have specific roles in the development of certain cell types. Although they are speculated to regulate gene expression in developmental contexts, a direct link between B-type lamins and developmental gene expression in an *in vivo* system is currently lacking. Here, we identify lamin B1 as a key regulator of gene expression required for the formation of functional olfactory sensory neurons. By using targeted knockout in olfactory epithelial stem cells in adult mice, we show that lamin B1 deficient neurons exhibit attenuated response to odour stimulation. This deficit can be explained by decreased expression of genes involved in mature neuron function, along with increased expression of genes atypical of the olfactory lineage. These results support that the broadly expressed lamin B1 regulates expression of a subset of genes involved in the differentiation of a specific cell type.

[1] Department of Biology, The Johns Hopkins University, Baltimore, Maryland 21218, USA. [2] Department of Embryology, Carnegie Institution for Science, Baltimore, Maryland 21218, USA. [3] Monell Chemical Senses Center, Philadelphia, Pennsylvania 19104, USA. [4] Department of Basic Medical Sciences, Neuroscience and Sensory Organs, University of Bari 'A. Moro', Bari 70121, Italy. [5] Department of Medicine, Molecular Biology Institute and Department of Human Genetics, University of California, Los Angeles, California 90095, USA. Correspondence and requests for materials should be addressed to Y.Z. (email: zheng@ciwemb.edu) or to H.Z. (email: hzhao@jhu.edu).

Lamins, nuclear intermediate filament proteins, are the major constituents of the nuclear lamina, a protein network under the nuclear envelope that functions in maintaining the structure and organization of the nucleus. Despite broad expression, lamins B1 and B2 appear to have specific roles in the development, differentiation and aging of certain tissues and cell types[1]. Lamins have speculated roles in regulating changes in and the long-term stabilization of gene expression during differentiation[2,3], but studies have not yet shown a direct link between B-type lamins and the expression of genes involved in development. Furthermore, these roles for B-type lamins have recently been challenged by reports that lamin B mutant embryonic stem cells do not exhibit deficits in gene expression or gene association with the nuclear lamina[4,5]. The lack of in depth in vivo mutational studies have left the question of how B-type lamins regulate development, differentiation and aging unanswered.

Deletion of the gene encoding lamin B1 (Lmnb1) in mice produces severe defects in the development of the nervous system, while many other tissues remain intact[4,6]. Lamin B1 has been shown to have diverse functions in many cellular processes in vitro and in invertebrates[2,3], and it remains unclear which functions of lamin B1 underlie the specific requirement in the nervous system. There have been reports that neuronal genes relocate to and from the nuclear lamina in correlation with changes in gene expression during differentiation or neuronal activation[7–14], yet evidence showing a direct role for lamins in the expression of these genes is lacking. Unfortunately, the perinatal lethality and extensive cell death observed upon Lmnb1 knockout in the embryonic nervous system has made in vivo analysis difficult.

We sought to investigate the role of lamin B1 in the development of neurons using the olfactory epithelium because it is a site of robust neurogenesis in adult animals. Resident stem and progenitor cells produce all neuronal and non-neuronal cell types of the epithelium throughout the lifespan of mammals in response to normal turnover or damage[15–17]. The differentiation of stem/progenitor cells into mature neurons involves characteristic changes in cellular morphology, connectivity and gene expression. The well-characterized neuronal differentiation program, the peripheral location and the robust, inducible neurogenesis make the olfactory epithelium an optimal system to study the role of lamin B1 in the development of neurons in adult animals.

We conditionally deleted Lmnb1 from a population of postnatally established, quiescent stem cells in the olfactory epithelium and examined the differentiation and function of olfactory sensory neurons lacking lamin B1. In the absence of lamin B1, olfactory sensory neurons exhibit attenuated responses to odour stimulation and abnormal nuclear pore distribution. Using a combination of candidate and unbiased profiling approaches, we show that this functional deficit is likely the result of decreased expression of specific genes that are required in mature olfactory sensory neurons.

## Results

### Lmnb1 conditional knockout in the adult olfactory epithelium.
We designed a genetic system to deplete Lmnb1 in the adult olfactory epithelium to avoid the perinatal lethality and wide spread cell death produced by lamin B1 knockout in the brain. Lmnb1 knockout was confined to horizontal basal cells, a population of postnatally established resident olfactory epithelium stem cells[16,18] by exploiting their expression of cytokeratin 5 (K5)[19]. Mice carrying a K5 promoter-driven Cre recombinase transgene (K5Cre[20]) were crossed to mice carrying a conditional Lmnb1 allele[21] (Lmnb1fl, Supplementary Fig. 1a). A Cre-dependent red fluorescent reporter allele[22] (R26Ai9) was introduced to allow for the visualization and isolation of mutant horizontal basal cells and their progeny based on tdTomato (Tomato) fluorescence (Supplementary Fig. 1a–c).

Under laboratory housing conditions, horizontal basal cells are largely quiescent[23,24]. Accordingly, olfactory epithelia from mutant animals (K5Cre;Lmnb1fl/fl) were almost entirely composed of Tomato-negative Lmnb1fl/fl cells, with the exception of Tomato-positive (presumptively Lmnb1−/−) horizontal basal cells (Fig. 1a, Supplementary Fig. 1c–d). Horizontal basal cells can produce all cell types of the olfactory epithelium upon tissue damage[16,18], allowing us to induce expansion of Lmnb1−/− cells through temporally controlled damage. We induced olfactory epithelium damage chemically using the drug methimazole (Supplementary Fig. 1e), which is known to activate horizontal basal cells[16]. Indeed, following regeneration, Tomato-positive cells (horizontal basal cells and progeny) were lamin B1-deficient in mosaic mutant olfactory epithelium based on antibody staining (Fig. 1b,c). By contrast, neighbouring Tomato-negative cells (Lmnb1fl/fl) exhibited normal lamin B1 staining (Fig 1b,c). Quantification of lamin B1 antibody staining revealed that 93.7% of all Tomato-positive cells were lamin B1 negative, compared to 3.5% of control cells (Supplementary Fig. 1d). Given the small minority (6.3%) of Tomato-positive cells expressing lamin B1, Tomato-positive cells in mosaic mutant epithelium will henceforth be referred to as Lmnb1−/−. Thus, recovery from injury produced a mosaic mutant olfactory epithelium, consisting of both control (Lmnb1fl/fl) and mutant (Lmnb1−/−) cells (Fig. 1; Supplementary Fig. 1).

Lmnb1 mosaic mutant olfactory epithelia were grossly indistinguishable from control epithelia by DIC microscopy or DAPI staining at all time points analysed (Supplementary Fig. 1f, see below). Examination of sparse Tomato-expressing cells in mosaic mutant tissue revealed Lmnb1−/− cells with the typical morphology of olfactory sensory neurons, stem cells, progenitors and supporting cells (Fig. 1d). Moreover, sparse Tomato-positive clones in mosaic mutant animals (Lmnb1−/− cells) and control littermates (Lmnb1+/− cells) consisted of similar proportions of sustentacular cells, neurons and progenitors (Supplementary Fig. 1g).

### Lmnb1 knockout produces attenuated olfactory neuron response.
Given that many Lmnb1−/− cells exhibited the typical bipolar morphology of mature neurons, we tested if olfactory sensory neuron function was affected by Lmnb1 knockout. We recorded the electrical responses of single, dissociated Lmnb1−/− olfactory sensory neurons to odour stimulation using the suction pipette technique[25]. Tomato-positive Lmnb1−/− (mutant) or Lmnb1+/− (control) olfactory neurons were dissociated from mosaic mutant or mosaic heterozygote epithelia, respectively, and stimulated with a two-odorant mixture (Cineole and Acetophenone). Fewer mutant neurons responded to odour stimulation, and, for neurons that did respond, the response amplitude was much smaller in mutant cells compared to controls (Fig. 1e–i). At 100 μM of each odorant, the highest concentration tested, 19% of mutant neurons responded, compared to 32% of control neurons, and the average response amplitude of mutant neurons was less than half of that in controls (Fig. 1e,g). Dose-response analysis revealed significant decreases in the average response amplitude of mutant cells at all concentrations above 10 μM of the odorant mixture (Fig. 1h,i). Moreover, the dose-response relation appeared flattened in mutant neurons (Fig. 1i), showing that lack of lamin

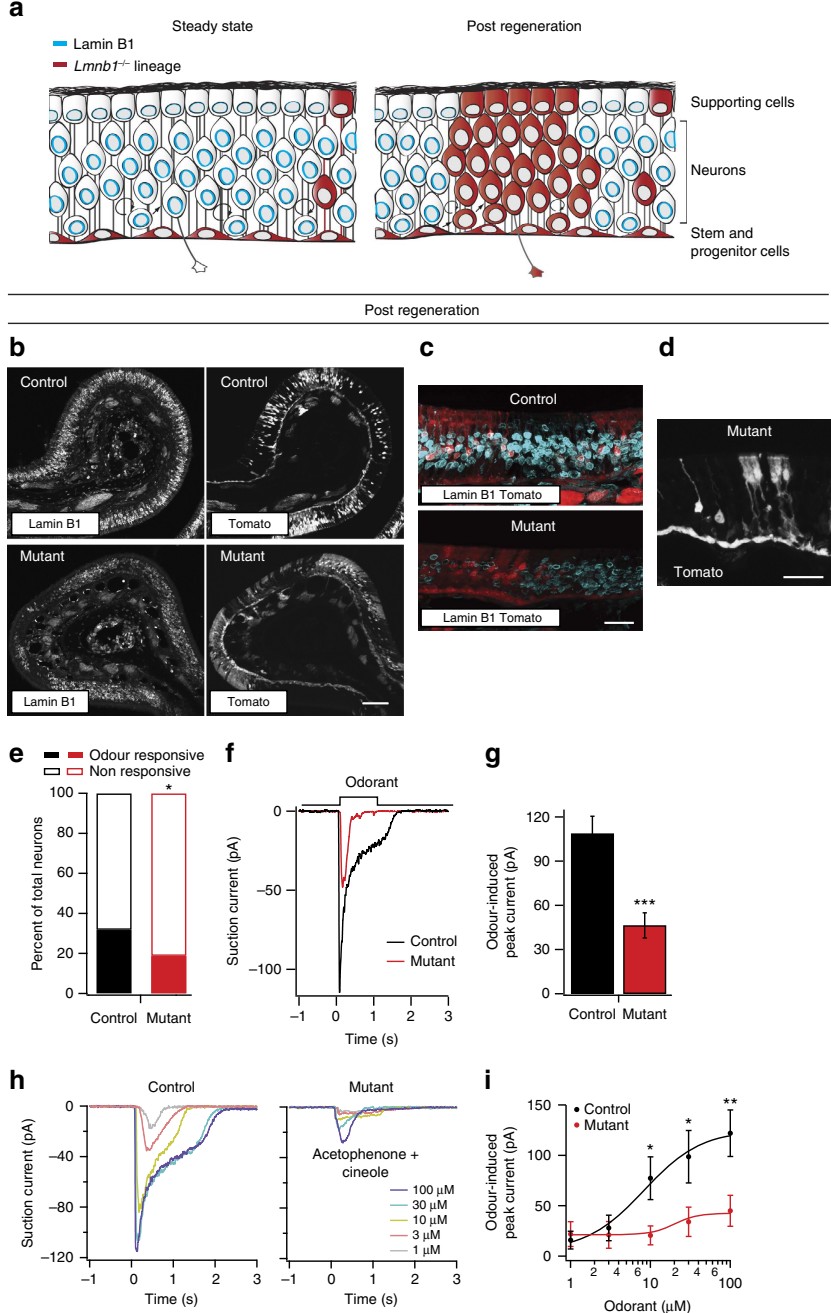

**Figure 1 | Mosaic knockout of *Lmnb1* in the olfactory epithelium and response of *Lmnb1*$^{-/-}$ neurons to odour stimulation. (a)** Cartoon depiction of *Lmnb1* mosaic knockout strategy during adult neurogenesis in the olfactory epithelium. Under normal laboratory conditions (Steady-State, left), Cre-expressing *Lmnb1*$^{-/-}$ horizontal basal cells (red) are largely quiescent and the majority of the olfactory epithelium expresses lamin B1 protein (blue). After damage-induced activation, *Lmnb1*$^{-/-}$ horizontal basal cells give rise to interspersed *Lmnb1*$^{-/-}$ daughter cells (red) that become all cell types of the epithelium, resulting in a mosaic epithelium containing both *Lmnb1*-null and *Lmnb1*-expressing cells (Post regeneration, right). **(b,c)** Lamin B1 antibody staining of control (*K5Cre;Lmnb1*$^{fl/+}$) and mosaic mutant (*K5Cre; Lmnb1*$^{fl/fl}$) olfactory epithelium post regeneration. Horizontal basal cell lineage can be identified by expression of a Cre-dependent Tomato reporter allele. Tomato-expressing cells are presumptively *Lmnb1*$^{-/-}$ in mutant epithelium, *Lmnb1*$^{+/-}$ in controls. Tomato-negative cells are presumptively *Lmnb1*$^{fl/fl}$ in mutant epithelium, *Lmnb1*$^{fl/+}$ in control. Scale bars indicate 50 µm **(b)** and 25 µm **(c)**. **(d)** Sparse Tomato expression in *Lmnb1*$^{-/-}$ cells in the mosaic mutant epithelium showing cellular morphology of mutant cells. Scale bar indicates 25 µm. **(e)** Percentage of odorant-responsive neurons. 32% (35 out of 110) of control (*Lmnb1*$^{+/-}$) neurons and 19% (23 out of 122) of mutant (*Lmnb1*$^{-/-}$) dissociated Tomato-positive neurons responded to a 1-s stimulation of 100 µM odorant mix (100 µM Acetophenone and 100 µM Cineole). $^{*}P < 0.05$, chi-squared test. **(f)** Representative responses of a control and a mutant neuron to a 1-s stimulation of 100 µM odorant mix. **(g)** Average peak response amplitude. Control neurons, 108.8 ± 11.7 pA ($n = 35$); mutant neurons, 46.4 ± 8.5 pA ($n = 23$). Data are expressed as mean ± s.e.m. $^{***}P < 0.001$, unpaired Student's *t*-test **(h)** Dose-response traces of a control and a mutant neuron to a 1-s stimulation of odorant mix at increasing concentration from 1 to 100 µM. **(i)** Dose-response relationships of the peak amplitude to the odorant mix. $n = 10$ cells. All data are expressed as mean ± s.e.m. $^{**}P < 0.01$, $^{*}P < 0.05$, unpaired Student's *t*-test.

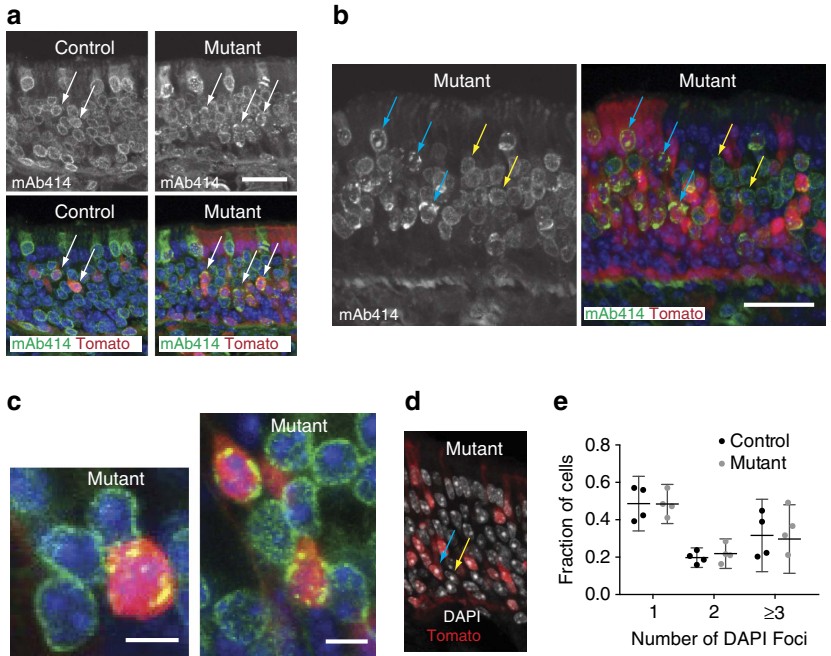

**Figure 2 | Nuclear architecture and organization of *Lmnb1*$^{-/-}$ cells.** (**a**) Distribution of nuclear pore complexes by antibody staining with mAb414, which recognizes several FG nucleoporins, in *Lmnb1* mosaic mutant and control epithelium. Arrows show nuclei of Tomato-positive cells, *Lmnb1*$^{-/-}$ in mutant and *Lmnb1*$^{+/-}$ in control. Scale bar indicates 25 μm. (**b,c**) Higher magnification of mAb414 staining in *Lmnb1* mosaic mutant epithelium. Blue arrows indicate nuclei of *Lmnb1*$^{-/-}$ cells; yellow arrows show *Lmnb1*$^{fl/fl}$ cells. Scale bars indicate 25 μm (**b**) and 5 μm (**c**). (**d**) Single confocal slice of DAPI stained nuclei from *Lmnb1* mosaic mutant olfactory epithelium showing DAPI-bright foci in the nucleoplasm indicative of heterochromatin reorganization. A single DAPI-bright focus can be seen in a Tomato positive *Lmnb1*$^{-/-}$ cell (blue arrow) and Tomato-negative *Lmnb1*$^{fl/fl}$ cell (yellow arrow). (**e**) Quantification of the number of DAPI bright foci in mutant (*Lmnb1*$^{-/-}$) and control (*Lmnb1*$^{fl/fl}$) cells. Error bars indicate ± 95% confidence intervals. Data are from four mice, >200 cells per mouse, two independent groups.

B1 disrupted the neurons' ability to dynamically encode odorant concentration. The decrease in both the number of odorant-responsive cells and the response size suggests a role for lamin B1 in the formation of functional olfactory sensory neurons.

**Lamin B1 is required for proper nuclear pore distribution.** Lamins are generally believed to maintain the structure and organization of the nuclear periphery[2], and several recent studies have underscored the importance of the nuclear envelope in the differentiation of neurons and other cell types[26]. We, therefore, examined the nuclei of *Lmnb1*$^{-/-}$ cells in the olfactory epithelia. Both gross nuclear shape and the expression pattern of lamin B2 appeared unchanged in *Lmnb1*$^{-/-}$ cells (Supplementary Fig. 2a–c). Despite normal staining pattern for lamin B2, antibody staining revealed clustered nuclear pore complexes in *Lmnb1*$^{-/-}$ cells, while nuclear pores were evenly distributed in neighbouring *Lmnb1*$^{fl/fl}$ cells in mosaic mutant tissue and in all cells in control tissue (Fig. 2a–c).

Mature olfactory sensory neurons exhibit a unique nuclear reorganization of peripheral heterochromatin to one or a few chromocenters in the nucleoplasm during differentiation, through a process involving the downregulation of Lamin B Receptor and lamin A/C[8,27]. *Lmnb1* knockout did not impact the formation of these chromocenters (Fig. 2d,e), which can be observed as DAPI bright foci enriched in HP1, H4K20me3 and H3K9me3 (ref. 8). Accordingly, cells in the neuronal layer of *Lmnb1* mosaic mutant olfactory epithelia were negative for LBR and lamin A/C staining (Supplementary Fig. 2d,e).

Altogether, these data show that lamin B1 is not required for all aspects of nuclear architecture in the olfactory neuron lineage but is necessary to maintain even nuclear pore distribution.

**Lmnb1 facilitates mature olfactory sensory neuron formation.** The abnormal response and nuclear pore distribution of *Lmnb1*$^{-/-}$ neurons prompted us to investigate any cellular and molecular changes that may underlie or result from these phenotypes. We employed a candidate approach to investigate all stages of olfactory neuron differentiation from stem cells.

Lamin B1 has been implicated in mitosis, cell cycle and cell death, yet neither proliferation rate nor number of apoptotic cells nor tissue thickness differed between mosaic mutant and control olfactory epithelia at any time point examined (Fig. 3a–c). Moreover, the expansion of *Lmnb1*$^{-/-}$ stem cells was comparable to control (*Lmnb1*$^{+/-}$) stem cells, based on histology and flow cytometry (Fig. 3d, Supplementary Fig. 3a,b). Regenerated *Lmnb1* mosaic mutant epithelia exhibited the stereotypical olfactory epithelium cellular organization. Sustentacular cells, olfactory progenitors and immature neurons were present in the mosaic mutant olfactory epithelium in a pattern similar to the control (Fig. 3e–i, Supplementary Fig. 3a–c). Many *Lmnb1*$^{-/-}$ cells expressed markers of neural progenitors (Sox2 and LSD1) and immature neurons (GAP43) (Fig. 3e–i). Moreover, *Lmnb1*$^{-/-}$ Tomato-positive nerves innervated glomeruli of the olfactory bulb (Supplementary Fig. 3d,e), the target of olfactory sensory neurons, suggesting that *Lmnb1*$^{-/-}$ cells were capable of becoming immature olfactory sensory neurons.

In contrast, *Lmnb1*$^{-/-}$ cells were either decreased in or devoid of olfactory marker protein (OMP), the most widely used marker of mature olfactory sensory neurons (Fig. 4a,b). This observation was confirmed by western blot of mosaic mutant olfactory mucosa, which revealed decreased expression of OMP and a second mature neuron protein, adenylyl cyclase 3 (AC3) in fully regenerated samples (Fig. 4c,d, Supplementary Fig. 4a,b), when the epithelium is composed of mostly mature neurons[16,28].

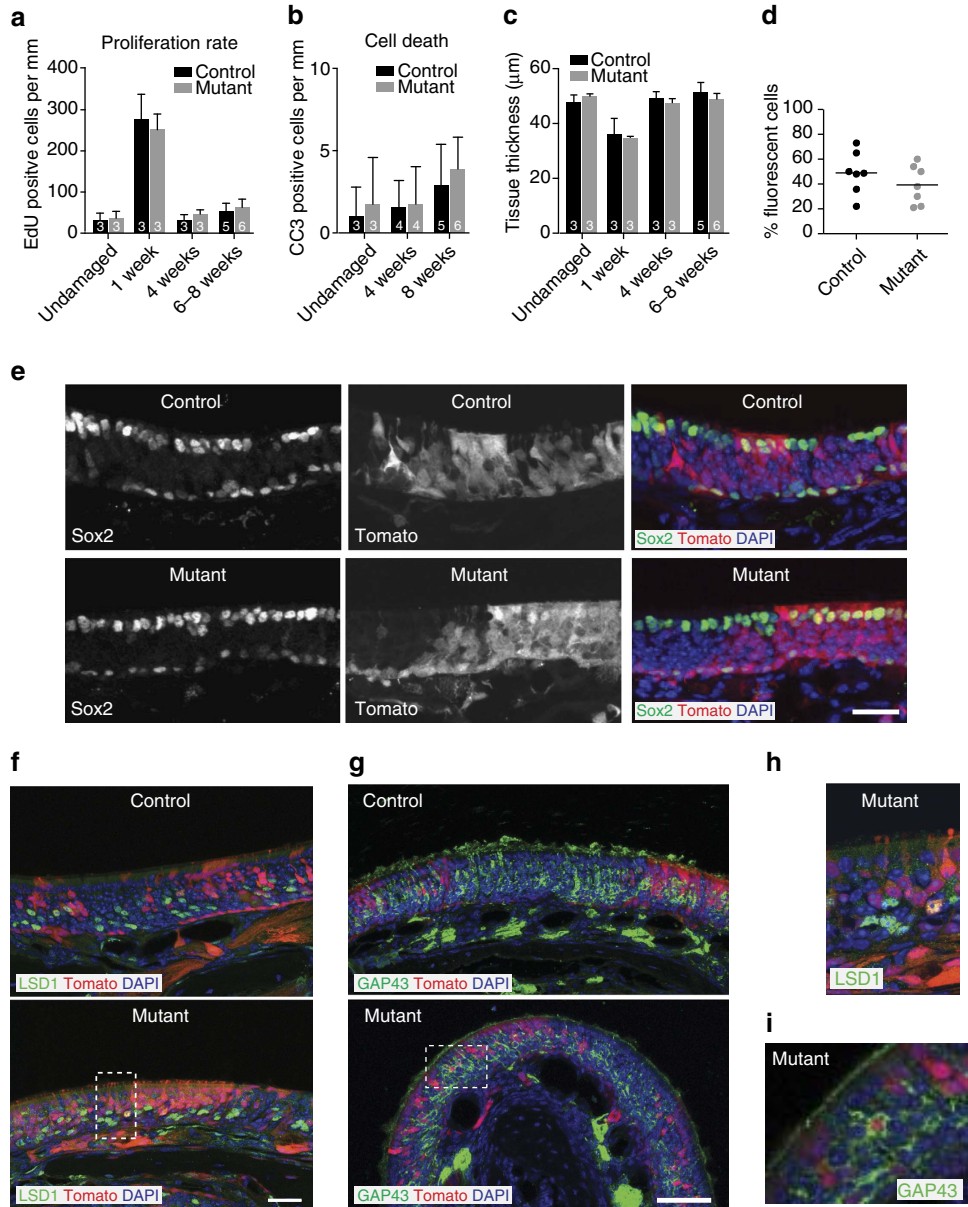

**Figure 3 | Cellular dynamics and distribution in the *Lmnb1* mosaic mutant olfactory epithelium.** (**a**) Proliferation rate in *Lmnb1* mosaic mutant and control olfactory epithelium. Cells retaining EdU after a 24-h pulse were counted in undamaged tissue (Steady-state, Fig. 1a) or at various time points after damage-induced regeneration. Mean + s.e.m. are displayed; number of samples (*n*) is reported on each bar; each *n* represents data from one animal. (**b**) Apoptosis rate in *Lmnb1* mosaic mutant olfactory epithelium. Cells that stained for cleaved caspase 3 (CC3) were counted in undamaged tissue (Steady-State) or at several time points after damage-induced regeneration. Mean + s.e.m. are shown; number of samples (*n*) is shown on each bar; each *n* represents data from one animal. (**c**) Average thickness of mosaic mutant olfactory epithelium. Mean + s.e.m. are shown; *n* is shown on each bar; each *n* represents data from one animal. (**d**) Percentage of Tomato-positive cells in mosaic mutant and control olfactory epithelium 4 weeks after methimazole-induced regeneration, as determined by flow cytometry. Each point represents data from one animal. (**e**) Expression of pluripotency marker Sox2 by antibody staining in the olfactory epithelium of mosaic mutant (*K5Cre;Lmnb1^{fl/fl}*) and control (*K5Cre;Lmnb1^{fl/+}*) mice. Sox2 is expressed in the nuclei of apical supporting cells and basal progenitors. Tomato-expressing cells are *Lmnb1^{−/−}* in mutants and *Lmnb1^{+/−}* in controls. Scale bar indicates 25 µm. (**f,h**) Antibody staining for progenitor marker, LSD1. Scale bar indicates 25 µm. (**g,i**) Expression of immature olfactory neuron protein, growth-associated protein of 43 kDa (GAP43) by antibody staining. Scale bar indicates 50 µm.

Mutant olfactory mucosa displayed decreased levels of lamin B1 in all samples that had undergone regeneration (Fig. 4c,d, Supplementary Fig. 4a,b), even though samples were prepared from mosaic tissue. In contrast, levels of GAP43, lamin B2 and actin were comparable to controls at all time points (Fig. 4c,d, Supplementary Fig. 4a,b). Taken together, these data suggest that *Lmnb1^{−/−}* cells are capable of becoming immature neurons but have reduced ability to become mature neurons.

One of the earliest events in olfactory sensory neuron maturation is odorant receptor expression[29], where each mature neuron expresses strictly one allele of one odorant receptor gene out of over 1,200 genes in the mouse genome[30–32]. We investigated the expression of an odorant receptor reporter allele, *M72-IRES-tauLacZ* (ref. 33) in *K5Cre;Lmnb1^{fl/fl}* mice. Whole mount X-gal staining of *Lmnb1* mosaic mutant olfactory epithelia revealed a significant decrease in the number of cells

expressing odorant receptor M72 4 weeks after regeneration (Fig. 4e,f, Supplementary Fig. 4c). Independently, the number of cells expressing a different odorant receptor reporter, *P2-IRES-tauLacZ* (ref. 34), was also decreased (Fig. 4f, Supplementary Fig. 4d). Zonal distribution (Fig. 4e) and axon targeting of either P2 or M72 olfactory neurons to the olfactory bulb was comparable to controls Given no evidence of increased cell death in mutant tissue (Fig. 3a–c), fewer cells expressing a given

odorant receptor could suggest altered odorant receptor expression. Altogether, these results support a requirement for lamin B1 in the formation of mature olfactory sensory neurons.

**Transcriptome analysis of *Lmnb1*$^{-/-}$ cells.** We observed decreased expression of several mature neuron proteins in mosaic *Lmnb1* mutant olfactory epithelium (Fig. 4, Supplementary

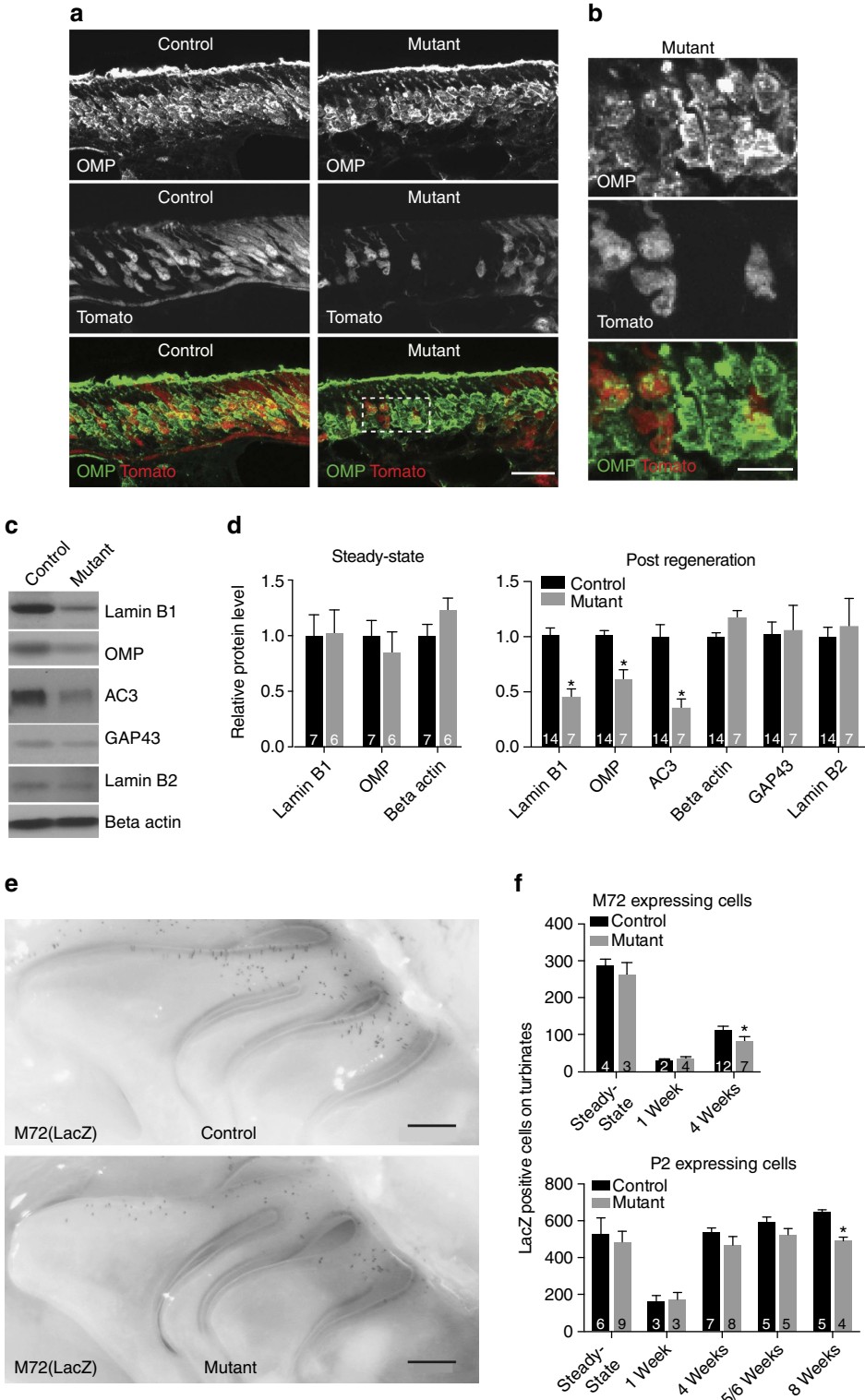

Fig. 4); however, we did not observe a compensatory increase in any other cell type, including immature neurons or neural progenitors (Fig. 3, Supplementary Fig. 3). This led us to take an unbiased profiling approach to investigate the molecular cause underlying the disruption in mature neurons upon $Lmnb1$ knockout.

$Lmnb1^{-/-}$ cells were isolated from mosaic mutant olfactory epithelia by fluorescence-activated cell sorting based on Tomato fluorescence 4 weeks after damage-induced regeneration (Supplementary Fig. 5a,b), the earliest time point at which decreases in mature neuron proteins were observed (Fig. 4, Supplementary Fig. 4). Sorted Tomato positive, $Lmnb1^{+/-}$ cells were used as controls. PCR of complementary DNA (cDNA) from sorted cells revealed $Lmnb1$ transcript depletion in cells sorted from mosaic mutant epithelia compared to control (Fig. 5a). RNA-seq was performed using RNA extracted from $>150,000$ sorted cells, resulting in $>50$ million 50 bp reads per sample (GSE80044). Differential expression analysis using edgeR (Bioconductor) revealed 626 genes significantly downregulated (Supplementary Data 1) and 185 genes upregulated (Supplementary Data 2) in $Lmnb1^{-/-}$ cells compared to $Lmnb1^{+/-}$ controls (FDR $< 0.05$) (Fig. 5b,c, Supplementary Fig. 5c). An independent ChIP-seq analysis of active promoter mark, H3K4me3, by FARP-ChIP-seq for low-cell number[35] revealed that changes in gene expression correlated with changes in the accumulation of H3K4me3 at promoter regions in sorted $Lmnb1^{-/-}$ cells (GSE80290, Fig. 5d,e, Supplementary Fig. 5d–g). Furthermore, the expression levels of several candidate genes determined by qPCR of lamin B1 mosaic mutant olfactory epithelium correlated with RNA-sequencing results (Supplementary Fig. 6a).

Gene ontology (GO) analysis of genes upregulated in $Lmnb1^{-/-}$ samples revealed three categories that reached statistical threshold (FDR $< 0.05$): lipocalin superfamily, cell cycle and extracellular matrix (Supplementary Data 3). 11 of the 185 upregulated genes were members of the lipocalin superfamily, associated with odorant binding (Supplementary Data 2). We also found several genes involved in immune response, cell death and DNA packaging (specifically Histone H1 variants). Many of the genes upregulated in $Lmnb1^{-/-}$ cells are not known to be highly expressed in any cell type of the olfactory epithelium, including several genes involved in diverse signalling pathways. For instance, $Lmnb1^{-/-}$ cells expressed high levels of g protein subunit $Gng5$, which is not typically expressed in olfactory neurons. One exception was the up regulation of Pax6, which is typically expressed in the olfactory epithelium in a subset of progenitors, supporting cells and duct cells[36]. The upregulation of genes not typically expressed in the olfactory neuron lineage supports the idea that $Lmnb1^{-/-}$ cells are not simply arrested in an immature state or becoming another olfactory epithelium cell type but may be acquiring an atypical fate. Alternatively, $Lmnb1^{-/-}$ cells may be arrested in a normally transient stage of development that has not yet been characterized.

Even though the sorted cells used for RNA-seq included stem cells, progenitors, neurons and supporting cells, GO analysis of genes downregulated in $Lmnb1^{-/-}$ cells revealed many processes specific to neurons (Supplementary Data 4). Many downregulated genes were specific to mature olfactory sensory neurons, including members of the olfactory signal transduction cascade (Fig. 5c). Comparison of changes in gene expression in $Lmnb1^{-/-}$ cells with published transcriptome datasets for individual olfactory epithelium cell types[37,38] revealed correlation between differentially expressed genes and genes expressed in mature neurons but not with genes expressed in other cell types of the epithelium (Supplementary Fig. 6b–i), supporting a specific deficit in the expression of mature olfactory neuron genes in $Lmnb1^{-/-}$ cells.

We next examined the expression of genes that are known to regulate olfactory sensory neuron maturation, including genes involved in genome reorganization[8,39], epigenetic changes[38,40], transcription factor expression[41], the unfolded protein response[42] and protein feedback[43–45]. A total of 27 odorant receptors were downregulated in $Lmnb1^{-/-}$ cells, including $Olfr17$ (P2), but the variation between samples for most odorant receptors was too high to make conclusions (Supplementary Fig. 6b). Genes involved in the unfolded protein response, cilia trafficking, axon targeting and activity-dependent feedback were downregulated in $Lmnb1^{-/-}$ cells, while genes involved in heterochromatin reorganization and epigenetic changes during neuronal maturation were unchanged. A total of 11 out of 35 genes involved in odorant receptor expression were differentially expressed in in $Lmnb1^{-/-}$ cells; out of 45 genes expressed in the nuclear lamina, zero were differentially expressed in $Lmnb1^{-/-}$ cells (gene lists can be found in Supplementary Data 5).

Enrichment of H3K9me3, a marker of constitutive heterochromatin, occurs at odorant receptor clusters during olfactory neuron differentiation and is thought to be involved in odorant receptor regulation and silencing[40,46]. To determine if lamin B1 affected the distribution of this marker at odorant receptor loci and across the genome, H3K9me3 FARP-ChIP-seq was performed on sorted Tomato-positive $Lmnb1^{-/-}$ and $Lmnb1^{+/-}$ cells. Precipitated DNA was sequenced producing $\sim 50$ million 50 bp reads per sample ($n = 2$, GSE80290). There were no changes in the distribution of H3K9me3 across the genome or at odorant receptor clusters in $Lmnb1^{-/-}$ cells (Fig. 5f,g, Supplementary Fig. 5h–i).

Transcriptome and ChIP analyses of $Lmnb1^{-/-}$ cells complemented the electrophysiological and protein expression findings, leading us to conclude that lamin B1 is dispensable for early differentiation in the olfactory sensory neuron lineage, but is required for the formation of functional mature neurons. Furthermore, the gene expression changes caused by $Lmnb1$

**Figure 4 | Formation of mature olfactory sensory neurons in _Lmnb1_ mosaic mutant olfactory epithelium.** (**a,b**) Antibody staining for mature olfactory sensory neuron marker, olfactory marker protein (OMP), in olfactory epithelium of control (_K5Cre;Lmnb1_$^{fl/+}$) and mosaic mutant (_K5Cre; Lmnb1_$^{fl/fl}$) mice. Tomato expression indicates $Lmnb1^{-/-}$ cells in mutant, $Lmnb1^{+/-}$ cells in control. Scale bars indicate 25 μm (**a**) and 10 μm (**b**). (**c**) Western blot of olfactory mucosa proteins from mosaic mutant (_K5Cre;Lmnb1_$^{fl/fl}$) and control (_Lmnb1_$^{fl/fl}$) mice 8 weeks after methimazole-induced regeneration. AC3, mature olfactory neuron protein adenylyl cyclase 3; GAP43, an immature neuron marker. (**d**) Quantification of western blots showing relative protein levels in mosaic mutant and control olfactory mucosa. Steady-state, undamaged tissue; Post regeneration, 6–8 weeks after methimazole-induced damage. All values, except beta actin, were normalized to beta actin for the same sample. All values are plotted relative to control. Data are expressed as mean + s.e.m. Number of samples (_n_) is shown on each bar; each sample was taken from one animal. $^*P < 0.05$ unpaired Student's _t_-test, corrected for multiple comparisons using Holm-Sidak test. (**e**) Whole mount X-gal staining of olfactory epithelium from mosaic mutant and control animals carrying a LacZ-tagged allele of odorant receptor M72 (_M72-IRES-tauLacZ_). Dark puncta on olfactory turbinates are indicative of cells expressing the tagged odorant receptor allele. Scale bars indicate 0.5 mm. (**f**) Quantification of odorant receptor reporter allele expression for _M72-IRES-tauLacZ_ or _P2-IRES-tauLacZ_. Total positive cells on medial surface of nasal turbinates were counted for each animal. Data are expressed as mean + s.e.m.; number of samples (_n_) is shown on each bar; each _n_ represents data from one animal $^*P < 0.05$, unpaired Student's _t_-test.

mutation suggest a role for lamin B1 in the upregulation of mature neuron-specific genes during differentiation.

## Discussion

The findings of this study establish a link between lamin B1 and *in vivo* developmental changes in gene expression. We report

distinct decreases in the expression of mature neuron-specific genes and upregulation of genes not typically highly expressed in the olfactory lineage. This aberrant gene expression likely underlies the disrupted function of *Lmnb1* mutant cells, where $Lmnb1^{-/-}$ neurons exhibited reduced, altered odorant response along with a decrease in the number of cells responsive to odour stimulation. Together, our data support a necessary role for lamin

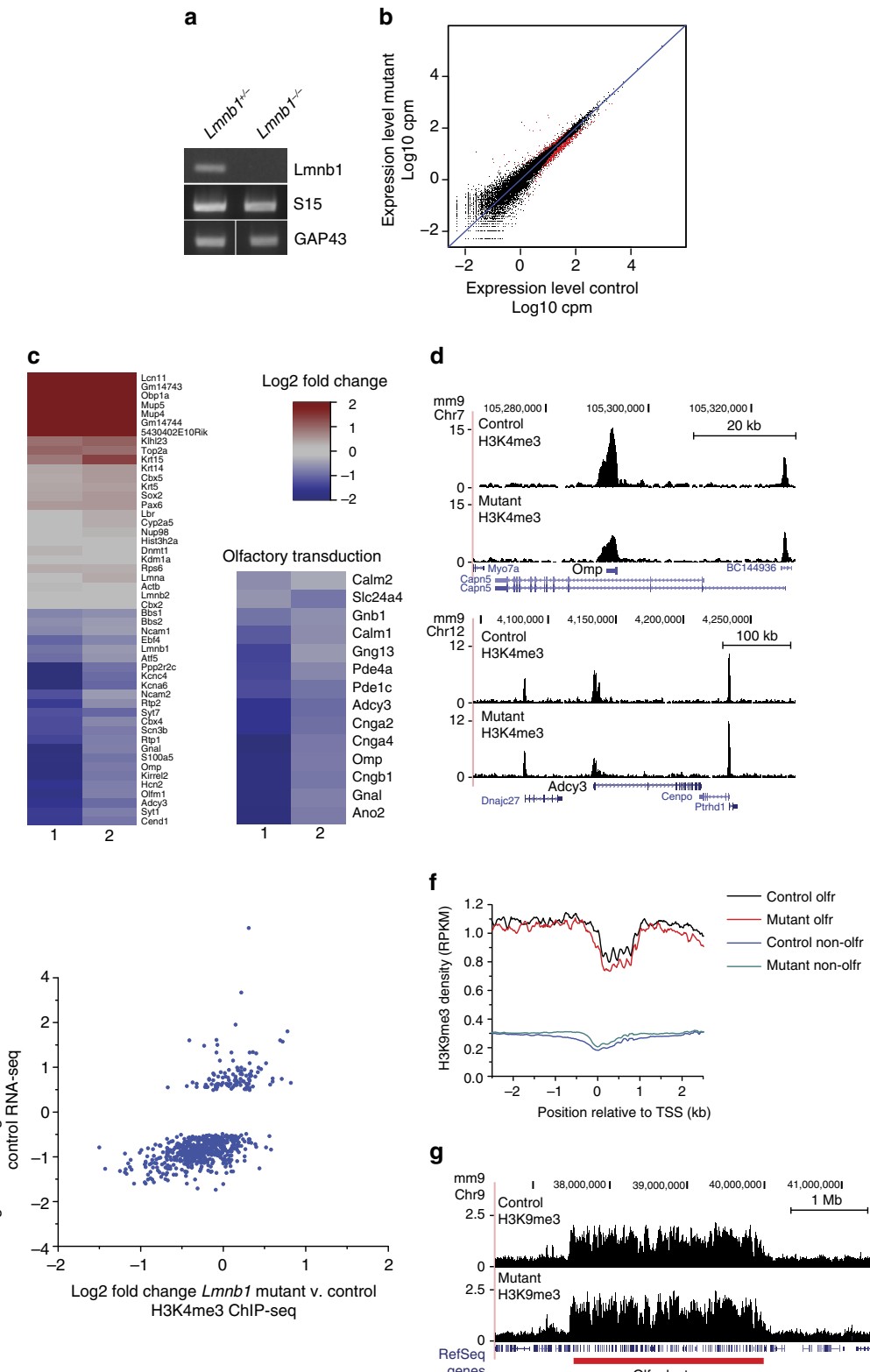

B1 in cell type-specific gene expression during development that is required for proper cell function.

Olfactory neuron differentiation involves extensive regulation to ensure that terminal differentiation does not proceed until a functional odorant receptor is expressed. Previous studies have shown that disrupting olfactory neuron differentiation results in decreases in mature neurons accompanied by accumulation of immature neurons and progenitors and/or increased cell death and epithelium thinning[8,41,42,47–49]. We report a novel phenotype upon *Lmnb1* mutation, whereby *Lmnb1*[−/−] cells progress through early stages of differentiation normally yet display defects at the final stage of olfactory sensory neuron maturation with no changes in cell death, tissue thinning or increase in any other cell type in the epithelium. The increased expression of genes not known to be highly expressed in any cell type of the olfactory epithelium may represent an arrest at a transient stage of neuronal development that is not typically observed, which may offer further opportunities to investigate olfactory neuron development.

We provide several lines of evidence that expression of at least a few odorant receptors is perturbed in *Lmnb1*[−/−] cells. Previous studies have suggested that the concerted action of heterochromatin inversion, epigenetic changes, transcription factor expression, the unfolded protein response and odorant receptor signalling ensure expression of exactly one functional odorant receptor (for review, see Monahan and Lomvardas[50] and Rodriguez[51]). We found evidence for normal heterochromatin inversion, epigenetic changes at odorant receptor genes, and expression of epigenetic modifiers in *Lmnb1*[−/−] cells, while observing decreased expression of transcription factors, members of the unfolded protein response, odorant receptor chaperones, odorant receptors and olfactory signal transduction proteins, including AC3, in *Lmnb1*[−/−] cells. A recent study suggests that AC3 is one of the earliest proteins expressed as part of a feedback loop that regulates odorant receptor expression[42,48]. Thus, the decreased levels of AC3 in *Lmnb1*[−/−] cells may result in the defects in odorant receptor expression we observed. Further investigation into odorant receptor regulation is needed to distinguish if *Lmnb1*[−/−] cells have exited the feedback loop[42,48], if *Lmnb1* mutation has revealed an additional layer of regulation, or if *Lmnb1* mutation results in defects upstream of odorant receptor expression during development.

The specific terminal differentiation phenotype produced by *Lmnb1* knockout in the olfactory epithelium is also in stark contrast to the cell cycle defects, massive cell death and improper migration observed upon *Lmnb1* knockout in the developing brain and retina[4,6,52]. Lamin B2 was the first B-type lamin implicated in nervous system development[53]; however, knockout of lamin B1 causes more severe neural defects[4,6,52]. Either targeted[6] or germ line[4] deletion of *Lmnb1* in mice produces widespread neural cell death, disrupted neuronal migration and decreased brain size late in embryonic development despite

normal brain patterning during early development, which may be caused by a defect in neuronal differentiation. In humans, *Lmnb1* duplications have been linked to demyelinating leukodystrophy[54], and *Lmnb1* polymorphisms have been linked to neural tube defects[55]. Future studies are needed to address if lamin B1 regulates gene expression during differentiation in other neuronal systems.

B-type lamins have been implicated in many cellular processes, so determining the role of an individual lamin in one process can be difficult, especially in the case of multiple severe phenotypes. The unique approach taken to knockout *Lmnb1* in the adult olfactory epithelium provides several advantages. In a mosaic knockout tissue, potential non-cell autonomous defects caused by *Lmnb1* knockout in a given cell could be compensated by nearby *Lmnb1*-expressing cells. Thus, our findings support at least a cell autonomous role for lamin B1 in olfactory neuron differentiation. In addition, as olfactory neuron differentiation in the adult olfactory epithelium mirrors differentiation during embryonic development in many ways, we expect that lamin B1 would play a similar role during embryonic development.

Disruption of B-type lamins has resulted in nuclear blebbing and/or disrupted nuclear pore organization in several cell types[6,21,56–59]. There is evidence that at least some aspects of lamin function, including nuclear pore distribution, may rely more on total lamin concentration than on a specific lamin type[57]. There is also evidence that lamin B1 and lamin B2 have distinct roles *in vivo* in the nervous system[4,60]. Mature olfactory sensory neurons do not express A-type lamins[8], and it is possible that the level of lamin B2 alone is insufficient to compensate the function of lamin B1. Consistent with this, we did observe disrupted distribution of nuclear pore complexes in *Lmnb1*[−/−] cells.

Nuclear pores are important in nuclear transport, cell signalling, genome organization and gene expression, and several recent studies have underscored the importance of nuclear pore complexes and nuclear transport in the regulation of cell fate and differentiation in neurons and other cell types[26,61]. In fact, B-type lamins have been shown to ensure nuclear retention of phosphorylated Erk by maintaining the even distribution of nucleoporins or nuclear pores in both *Drosophila*[58] and cultured mouse neurons[62]. It is, therefore, possible that abnormal nuclear pore distribution may underlie the changes in gene expression observed upon *Lmnb1* mutation in the olfactory lineage.

In general, B-type lamins are thought to regulate gene expression through repression of genes at the nuclear periphery[63,64]. This repressive role of lamins is based on observations that regions of the genome associated with the nuclear lamina exhibit low gene density, low expression and epigenetic signatures of hetero-chromatin; that experimentally targeting genes to the nuclear lamina can, though does not always, result in gene repression; and that lamin depletion has resulted in de-repression of silent genes in some cases[2,3]. Mature olfactory sensory neurons, however, exhibit a unique organization of most heterochromatin away from the

**Figure 5 | Transcriptome profiling of *Lmnb1*[−/−] mutant olfactory cells.** (**a**) PCR amplification of cDNA from sorted mutant (*Lmnb1*[−/−]) or control (*Lmnb1*[+/−]) Tomato positive cells. Thirty five cycles of PCR were performed to amplify *Lmnb1* exon 10/11, ribosomal protein S15 and immature neuron marker GAP43. (**b**) Scatterplot depicting expression level (average CPM) of genes in sorted mutant (*Lmnb1*[−/−]) versus control (*Lmnb1*[+/−]) cells (n = 2 independent experiments) based on RNA-seq. Each point represents one gene. Results of differential expression analysis are shown by colour: red FDR < 0.05; black FDR > 0.05. (**c**) Heatmaps of log2 fold change in expression of select genes in mutant and control cells. Genes involved in olfactory signal transduction are displayed separately. Data shown are for two experimental replicates (1 and 2). (**d**) Histograms showing the distribution of activate promoter histone modification, H3K4me3, across the locus of two mature olfactory neuron genes (*Omp* and *Adcy3* (AC3)) in sorted mutant and control cells based on FARP-ChIP-seq. (**e**) Correlation between changes in gene expression (log2 fold change from RNA-seq analysis) and H3K4me3 promoter abundance (log2 fold change from FARP-ChIP-seq analysis) in mutant cells compared to controls for genes differentially expressed by RNA-seq. Pearson correlation R = 0.55, P < 0.001 F-test. (**f**) Average distribution of constitutive heterochromatin mark H3K9me3 across odorant receptor (*olfr*) and non-odorant receptor genes (non-*olfr*) in *Lmnb1*[−/−] and *Lmnb1*[+/−] control cells based on FARP-ChIP-seq (n = 2 independent groups). (**g**) Representative histograms showing the distribution of H3K9me3 across an odorant receptor cluster in mutant and control cells.

nuclear periphery[8,65], where it is unlikely to be in direct contact with lamin B1. In contrast to being a general repressor of gene expression, we find that lamin B1 is required for the upregulation of lineage-specific genes involved in mature neuron function. Our observations, therefore, suggest that lamin B1 may play a broader role in gene expression than has previously been appreciated.

Gene association with the nuclear lamina can be rather dynamic, and several neuronal genes relocate to and from the nuclear lamina in correlation with changes in gene expression during differentiation or neuronal activation[7–9,12]. Lamin B1 association with genomic loci is most frequently determined by identifying regions of the genome methylated by a lamin B1-DNA adenine methyltransferase fusion protein (DamID)[66]; however, it is currently not feasible to perform DamID in olfactory sensory neurons due to the inability to culture them. Expression of genes that associate with lamin B1 in several tissue culture cells[7] seems unaffected by Lmnb1 knockout in olfactory sensory neurons (Supplementary Fig. 6k–n). Future technological advances that allow for the global analysis of chromatin-lamin association during olfactory neuron differentiation will shed light on the whether lamin B1 affects locus-specific organization or some other aspect of genome organization during olfactory sensory neuron differentiation.

Lastly, our examination of the olfactory epithelium revealed high expression of Lmnb1 in mature olfactory neurons, leading us to wonder if lamin B1 plays a role in the maintenance of mature, postmitotic neurons of the olfactory epithelium. To address this question, we attempted to use the OMP-Cre mouse line[67] to knockout Lmnb1 in mature olfactory sensory neurons (Supplementary Fig. 7). We observed clear evidence of recombination of genomic DNA in the olfactory epithelium of the OMP-Cre-driven conditional Lmnb1 mutants ($Omp^{Cre/+};Lmnb1^{fl/fl}$), but saw no changes in lamin B1 antibody staining in mutant olfactory epithelium (Supplementary Fig. 7a–c). We further found no difference in the number of OMP-positive mature neurons, tissue thickness, cell death or proliferation in OMP-Cre-driven conditional Lmnb1 mutants (Supplementary Fig. 7d–g). We did not observe any evidence of lamin B1 protein depletion in the olfactory epithelium of OMP-Cre-driven knockout mice, even after six months of age. The lack of lamin B1 protein depletion in $Omp^{Cre/+};Lmnb1^{fl/fl}$ olfactory epithelium could arise from inefficient recombination, despite the evidence of recombination that we observed. Alternatively, lamin B1 protein may be very stable in postmitotic neurons, which is supported by several lines of evidence in the literature. First, a peptide-labelling screen identified lamin B1 as one of the most stable proteins in neurons of the adult brain[68]. Another group has identified lamin B1 as a very stable protein in the retina[52]. Second, to date, all published strategies to knock out Lmnb1 or Lmnb2 in vivo in mice have been targeted to stem cells or other proliferative cell populations. Employing this strategy, B-type lamin knockout has been successful in the whole embryo[4,53,56], keratinocytes of the skin[21], progenitors of the forebrain[6], and stem cells in the olfactory epithelium (this study). Still, it remains unclear whether the failure to deplete lamin B1 in mature olfactory neurons was caused by lamin stability, incomplete recombination, or both. Regardless the reason, lamin B1 protein was not depleted in the $Omp^{Cre/+};Lmnb1^{fl/fl}$ line, leaving the role of Lmnb1 in the maintenance of mature, postmitotic neuron maintenance currently unclear.

## Methods

**Mice.** Mice were housed in 12-h light/dark cycle with access to food and water ad libitum. All procedures relating to mouse care and treatment were approved by and performed in accordance with the guidelines of the Animal Care and Use Committees of Johns Hopkins University and Monell Chemical Senses Center. Female and male mice maintained on a mixed genetic background aged 2–6 months were used in all assays. Mutant animals and littermate controls were assigned to the same treatment group; experimental methods and analyses were randomly assigned to the entire treatment group. All data collection was performed using numeric animal identifiers; animal genotype was later decoded and used for data analysis.

$Lmnb1^{fl}$ mice contain Lmnb1 exon 2 flanked by lox sites, resulting in a null allele upon Cre-mediated recombination[21] (Supplementary Fig. 1a). K5Cre mice contain a transgene for Cre recombinase under the Keratin 5 promoter[20]. Within the olfactory epithelium, K5 expression is limited to the horizontal basal cells; however, K5 is abundant in the epidermis, yet Lmnb1 knockout does not produce a phenotype in the epidermis[21]. Mice containing a Cre-dependent red fluorescent reporter allele ($Rosa^{tdTomato}$, $R26^{Ai9}$) were obtained from Jackson Laboratories[22]. P2-IRES-tauLacZ and M72-IRES-tauLacZ mice contain knock-in alleles of a fusion tau-LacZ downstream of an internal ribosome entry site following the coding sequence of the endogenous P2 or M72 odorant receptor gene[33,34]. $Omp^{Cre}$ mice contain Cre recombinase knocked in at the endogenous Omp locus[67]. To reduce the potential influence of strain background, all K5Cre, $Omp^{Cre}$, and $Lmnb1^{fl}$ mice were backcrossed to the B6 background for at least five generations.

**Stem cell activation through methimazole-induced damage.** Mice 2–3 months of age were given two 50 mg per kg intraperitoneal injections of methimazole (Sigma M8506) three days apart[28]. Mice recovered 1–8 weeks after the second injection before tissue collection (Supplementary Fig. 1C).

**Antibody staining.** All procedures were performed following standard protocol. Mice were transcardially perfused with PBS followed by 4% paraformaldehyde solution. Heads were postfixed in 4% paraformaldehyde, decalcified in 250 mM EDTA PBS at 4 °C over 3–6 days, cryoprotected in 30% sucrose PBS 4 °C for 3–6 additional days, then frozen in Tissue-Tek O.C.T compound and stored at −20 °C until cryosectioning, except for lamin A/C staining, which was performed on unfixed tissue sections that were post-fixed in methanol-acetone. 10 μm thick sections were subjected to immunofluorescence following standard protocol. All antibodies and dilutions are described in Table 1. Slides were mounted in Fluoromount and imaged on a Leica Sp5 confocal microscope. Tomato signal is endogenous fluorescence. All control images are from littermates in the same treatment group as the mutant animal. Reported cell death rate (cleaved caspase three positive cells per mm) for each animal ($n$) is an average of at least 15 fields of view taken from similar olfactory regions; ∼10 mm of tissue was counted for each animal. Tissue thickness for a given animal ($n$) was determined using at least 10 fields of view; similar areas of epithelium were examined in each animal.

For lamin B1 knockout quantification, cells in the neuron layer of the olfactory epithelium were counted as either positive or negative for lamin B1 antibody staining. All cells within a given field were counted. Olfactory epithelia were from $K5Cre;Lmnb1^{fl/fl}$ (mosaic mutant) and $K5Cre;Lmnb1^{fl/+}$ (mosaic control) littermate mice from the same treatment group. Data shown are from five sets of littermate mice; over 200 cells were counted for each mouse.

The numbers of different cell types produced by control ($Lmnb1^{+/−}$) and mutant ($Lmnb1^{−/−}$) horizontal basal cells were determined by the morphology and tissue stratification of Tomato positive cells within clones after tissue regeneration. Rectangular cells at the apical surface with basal nuclei were counted as sustentacular cells, tear shaped cells with processes that were located in the middle layer of the epithelium were counted as neurons, and rounded cells with no obvious processes at the base of the epithelium were counted as progenitors. Bowman's gland cells and other minor cell types were ignored in this analysis due to their rarity. Per cent of cells considers only those cell types that were counted. Olfactory epithelia were from $K5Cre;Lmnb1^{fl/fl}$ (mosaic mutant) and $K5Cre;Lmnb1^{fl/+}$ (mosaic control) mice. Data shown are from five sets of littermate mice; over 100 cells were counted for each mouse; images used in this study were also used in the lamin B1 knockout quantification described above. Graphing and statistical analyses were performed in Prism 6.

**Western blotting.** All procedures were performed following standard protocol. Whole olfactory mucosa was homogenized after mice were transcardially perfused with PBS. Protein estimations were performed using Pierce BCA Assay Kit (Life Technologies). Total protein of 25–50 μg was used for western blotting following standard protocol. Each sample ($n$) was taken from a single mouse; each sample was tested for all proteins; littermate controls that were part of the same treatment group were used for all mutant animals. Antibodies and dilutions are described in Table 1. Blots were developed using Perkin Elmer ECL Plus Lightning substrate (NEL105001EA), exposed to X-ray film and converted to scanned images. Relative protein abundance was determined by pixel intensity using ImageJ (NIH). Raw value for each band was normalized to beta actin for the same sample then reported relative to control average for each experimental group (littermates, same blot). Group normalization was performed to allow for comparison across different time points and account for any differences due to age and treatment group. Western blot images have been cropped for presentation, but full size images are presented in Supplementary Fig. 8 Graphing and statistical analyses were performed in Prism 6.

**Table 1 | Antibodies used in this study.**

| Antigen/reagent | Dilution (WB) | Dilution (IF) | Dilution (ChIP) | Source | Provider | Catalogue # or citation |
|---|---|---|---|---|---|---|
| Sox2 | NA | 1:400 | NA | Goat | Santa Cruz | SC-17320 |
| LSD1 | NA | 1:200 | NA | Rabbit | Abcam | ab17721 |
| Lamin A/C | NA | 1:5 | NA | Mouse | H. Herrmann | Kolb Nucleus 2011 |
| Lamin B Receptor | NA | 1:100 | NA | Guinea Pig | H. Herrmann | Hoffmann Nature Genetics 2002 |
| Cleaved Caspase 3 | NA | 1:400 | NA | Rabbit | Cell Signaling | 9661s |
| Phalloidin-AlexaFluor488 | NA | 1:400 | NA | NA | Invitrogen | A12379 |
| mAb414 | NA | 1:100 | NA | Mouse | Abcam | ab24609 |
| Lamin B1 | 1:2,000 | 1:200 | NA | Goat | Santa Cruz | SC-6216 |
| OMP | 1:10,000 | 1:1,000 | NA | Goat | Wako | 544-10001-WAKO |
| GAP43 | 1:5,000 | 1:400 | NA | Rabbit | Abcam | ab137910 |
| Lamin B2 | 1:2,500 | 1:100 | NA | Mouse | Thermo-Fisher | 33-2100 |
| AC3 | 1:10,000 | NA | NA | Rabbit | Santa-Cruz | SC-588 |
| Beta Actin | 1:5,000 | NA | NA | Rabbit | Cell Signaling | 4970L |
| H3K9me3 | NA | NA | $1\,\mu g\,ml^{-1}$ | Rabbit | Abcam | ab8898 |
| H3K4me3 | NA | NA | $0.5\,\mu g\,ml^{-1}$ | Rabbit | Cell Signaling | 9751S |

**EdU labelling.** All procedures were performed following standard protocol. Mice were injected IP with 500 μl of $12.5\,mg\,ml^{-1}$ EdU 24 h before tissue collection. EdU visualization was performed using Click-IT EdU labelling kit (Life Technologies C10337). Proliferation rate (EdU positive cells per mm) reported for a given animal ($n$) is an average of at least 12 fields of view. Fields of view were taken from similar olfactory regions across animals. Approximately 6 mm of tissue was counted for each animal. Graphing and statistical analyses were performed in Prism 6.

**Whole mount X-gal staining.** All procedures were performed following standard protocol. Mice were perfused with PBS then exposed turbinates of bisected heads were subjected to whole mount X-gal staining following standard protocol[33]. Olfactory epithelia were imaged using a stereomicroscope under low magnification. Values reported for each animal ($n$) represent the positive cells on turbinates as an average of left and right sides, except where tissue was damaged. Data collected for each of the two tagged odorant receptor alleles are from two independent cohorts of animals. Graphing and statistical analyses were performed in Prism 6.

**Counting nuclear chromocenters.** Number of DAPI-bright foci were counted for cells in the neuronal layer of the olfactory epithelium by analysing confocal Z-stacks of DAPI stained 10 μm cryosections from mosaic mutant animals containing Tomato reporter allele. Z stacks spanned 8 μm in Z direction. Nuclei were counted only if the entire nucleus was included in Z stack. Over 150 cells were counted for each of four animals from three or more fields of view. All animals were $K5Cre;Lmnb1^{fl/fl}$ genotype. Graphing and statistical analyses were performed in Prism 6.

**Single cell electrophysiology.** Responses of isolated olfactory sensory neurons were recorded using the suction-pipette technique[25]. Briefly, the cell body of an isolated olfactory sensory neuron was gently sucked into the tip of the recording pipette so that the cilia remain exposed to the bath solution. The recorded current (termed suction current), when filtered at DC—5,000 Hz ($-3\,dB$, 8-pole Bessel filter) represents the slow transduction current which enters at the cilia with superimposed fast capacitive currents corresponding to action potential firing at the onset of the current response. Fast solution changes and odorant exposures were achieved by transferring the tip of the recording pipette containing the neuron across the interface of neighbouring streams of solutions using the Perfusion Fast-Step solution changer (Warner Instrument Corporation). The receptor current was isolated by filtering the suction current with a bandwidth of DC-50 Hz ($-3\,dB$, 8-pole Bessel filter). The suction current was sampled at 10 kHz and recorded with a Warner PC-501A patch clamp amplifier, digitized using Power1401 II A/D converter and Signal acquisition software (Cambridge Electronic Design, U.K.).

Mammalian Ringer's solution contained (in mM) 140 NaCl, 5 KCl, 1 $MgCl_2$, 2 $CaCl_2$, 0.01 EDTA, 10 HEPES and 10 glucose. The pH was adjusted to 7.5 with NaOH. Acetophenone and Cineole 1 mM stock solution in Ringer's were prepared and the odorant solutions at their final concentration were prepared daily. All chemicals were purchased from Sigma-Aldrich. Fast solution changes and odorant exposures were achieved by transferring the tip of the recording pipette containing the neuron across the interface of neighbouring streams of solutions using the Perfusion Fast-Step solution changer (Warner Instrument Corporation). All experiments were performed at 37 °C.

**FACS sorting.** Olfactory mucosa was dissected into sterile PBS, cut into small pieces and then digested in 10 U Papain in 2.5 mM Cystein per 0.5 mM EDTA PBS at 37 °C for 15 min. Samples were triturated and treated with DNaseI

($\sim10$ U, Qiagen 79254) for 10 min at room temperature before being transferred to a 1% FBS PBS solution. Single cells were isolated using a cell strainer and sorted using a FACS Aria II (BD Biosciences). Gating was optimized using cells from $K5Cre;R26^{Ai9/+}$ mice after methimazole-induced regeneration and nonfluorescent control. Sorted cells were used in RNA-seq and ChIP-seq experiments.

**RNA sequencing and analysis.** Approximately 150,000 to 600,000 Tomato-positive mutant or heterozygous olfactory epithelial cells were sorted from a single mouse for each sample. Data are based on two sets of age-, sex- and treatment-matched littermates (four samples from four mice, two groups). Libraries were prepared using Illumina RNA Prep Kit 2 and sequenced, producing approximately 50 million 50 bp reads per sample. For detailed description, please see Supplementary Methods.

**ChIP sequencing.** Approximately 500,000 Tomato-positive mutant or heterozygous olfactory epithelial cells were sorted from a single mouse. Each sample was crosslinked, sonicated, split into two groups and subjected to ChIP with either anti-H3K9me3 or anti-H3K4me3 (Table 1). Data for H3K9me3 are based on two sets of age-, sex- and treatment-matched littermates; data for H3K4me3 are based on a single set. Libraries were prepared from precipitated DNA and sequenced, producing $\sim50$ million 50 bp reads per sample. For detailed description of analysis, please see Supplementary Methods.

**qPCR.** RNA was extracted from olfactory mucosa using TRIzol (Life Technologies) followed by DNase I digestion, according to the manufacturers' protocols. cDNA was generated using the RETROscript Reverse Transcription Kit (Thermo Fisher Scientific) with Oligo dTs. Primer sequences can be found in Supplementary Table 1. qPCR was performed on a StepOnePlus Real-Time PCR system (Applied Biosystems) using Maxima SYBR green/ROX qPCR Master Mix $\times2$ (Thermo Fisher Scientific). All reactions were performed in triplicate; Ct values were averaged for each gene in each sample. Results were analysed by the $2-\Delta\Delta Ct$ method[69] with normalization to the geometric mean of Actb, Gapdh and Ubc[70]. For details, see Supplementary Methods.

**Data availability.** Raw sequencing data and count data is available through the following GEO accessions: GSE80044 for RNA-seq data and GSE80290 for ChIP-seq data. In addition, all relevant data are available from the authors.

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

## Acknowledgements

We thank Y. Guo and L. Hugendubler for help with flow cytometry and F. Tan and A. Pinder for help with sequencing and analysis. We are thankful to H. Herrmann for the Lamin A/C and Lamin B Receptor antibodies. We thank R. Johnston, X. Chen, R. Kuruvilla and S. Hattar for critical discussion. We also thank members of the Zhao, Zheng, Hattar and Kuruvilla labs for suggestions and discussion. This work was supported by NIH grants DC007395 (H.Z.), GM056312 (Y.Z.), GM110151 (Y.Z.), AG035626 (S.G.Y.) and G20OD020296 (infrastructure improvement at the Monell Chemical Senses Center) and Ellison Medical Foundation (Y.Z.). C.M.G., F.N.D., X.Y. were partially supported by NIH training grant T32GM007231.

## Author contributions

C.M.G., Y.Z. and H.Z. designed experiments and wrote the manuscript. C.M.G. performed all experiments and analyses except electrophysiology, qPCR, and ChIP-seq analysis. M.D. and J.R. performed electrophysiological experiments and analyses. F.N.D. performed qPCR experiments and analysis. S.Y. aided in ChIP. X.Z. performed ChIP-seq sequence analysis. S.G.Y. provided *Lmnb1*$^{fl}$ mice. M.D., X.Z., S.Y., J.R. and S.G.Y. commented on the manuscript.

## Additional information

**Competing interests:** The authors declare no competing financial interests.

