## [Peer Review File · Nature Communications]

Reviewers' comments:

Reviewer #1 (Remarks to the Author):

This is an interesting manuscript seeking to characterize the role of lamin b1 in the development of olfactory neurons. The authors use a conditional laminb1 KO to delete this gene in horizontal basal cells of the olfactory epithelium, which are considered the quiescent stem cells of this tissue. By pharmacological ablation and of the neurons of the olfactory epithelium the authors promote the differentiation of these cells into neurons, and using a Cre- inducible fluorescent reporter they analyze specifically the KO cells. They find that mature markers and several components of olfactory signaling are downregulated in the KO neurons, explaining reduced electrophysiological responses to odorants. The experiments are rigorous and well executed and the findings very interesting since developmental roles for lamin b1 have not been previously described. Clearly, the manuscript lacks mechanistic insight into how lamin b1 regulates the expression of these genes and the intriguing demonstration of the altered distribution of nuclear pores does not really answer this. I believe that the observation is significant enough that merits publication without mechanistic experiments, however I would like to see a couple of additional experiments that would clarify the developmental role of laminb1. First, it would be very insightful to delete laminb1 with a mature driver, like OMP-Cre and ask if laminb1 is required for the maintenance of the mature olfactory neuron state. Second, the authors show a reduction of olfactory receptors M72 and P2 expression at later stages of HBC differentiation. I think this results points out to unstable expression of these two genes, which would be the consequence of Adcy3 downregulation (previous reports have shown that Adcy3 is required for stable OR expression). Thus, the authors should also attempt to delete laminb1 also with an ORiresCre line and measure the stability of OR expression in the absence of laminb1. I understand that the second experiment will take time to complete and analyze, especially if the authors do not have these mice at hand. However, it is imperative to at least entertain this possibility in the discussion of the paper. Finally, a minor point is that a lot of the genes that the authors find upregulated in their KO cells are specifically expressed in the Mash1 GBC population (our own unpublished data), show they should tone down the claim that these are genes that are not expressed normally in the olfactory epithelium and entertain the possibility that these are genes that are only transiently expressed during neuronal differentiation.

Reviewer #2 (Remarks to the Author):

The nuclear lamina is a network of proteins that regulates nuclear structure and function and consists of a two-dimensional matrix of mainly A- and B-type lamin proteins located beneath the inner nuclear membrane. Evidence from mice supports an important role for the B-type lamins (lamins B1 and B2) in neuronal migration and brain development. Lamin B1 has been implicated as a susceptibility gene in neural tube defects, and its duplication causes an adult onset neurodegenerative disorder in humans. Recent studies have also suggested that the expression of B-type lamins in somatic cells influences cellular senescence. It is unclear which functions of lamin B1 underlie its specific requirement in the developing nervous system. To study the possible role of lamin B1 in the postnatal development of neurons, the authors have used the olfactory epithelium because it is a site of neurogenesis in adult animals. Lamin B1 was knocked-out in mice in a subset of olfactory epithelial stem cells and its effects on differentiation and function of olfactory neurons were analyzed.

This is a potentially interesting paper that addresses an important issue using engineered mice; what happens with adult neurogenesis in the absence of lamin B1, does this change the function of the olfactory neurons and what happens with their gene expression.

However, I have several concerns, related to technical and/or interpretational problems, with the

study:

1. Cre is driven by K5, which is active during embryogenesis and in postnatal tissue. This system would remove lamin B1 from olfactory epithelial stem cells and differentiated neuronal cells during normal neurogenesis and acute regeneration. This removal should be indicated in the cartoon in Fig.1A for the steady-state, with a few of the neurons colored red. Depending on the age of the mice being analyzed, normal neurogenesis occurs in at least 2 waves, one during the first postnatal week and the second at 4 months. Is it possible these waves would be evident as reduced lamin B1 in the western for the steady-state sample group compared to control mice with intact lamin B1 alleles? What is the support for intact lamin B1 alleles during embryonic development of the olfactory epithelia in their system?

2. The tomato, lamin B1, and cre transgenes are from different constructs and strain backgrounds, which may result in variation in their sensitivity to the cre recombinase. Quantification of the overlap between tomato red positive cells and lamin B1 negative cells is necessary. This could be done by counting cells in sections that have been stained for lamin B1 and tomato, as seen in Fig. 1B. In the manuscript, the authors refer to tomato and lamin B1^{-/-} cells interchangeably (for example in Fig.1D, 2A, 2B, 2C). Even with the inclusion of such quantification, this is misleading and should be avoided.

3. In the odor stimulation experiments, and also in other experiments, the authors used the LaminB1^{+/-} as the control sample group for the experiments. The inclusion of a control with both Lamin B1 alleles intact would allow for the assessment of effects from haploinsufficiency and establish the normal baseline for the different experiments.

4. Further information regarding samples processed for RNA and chip sequencing would facilitate the interpretation of results. Specifically, number of samples as well as gender and age.

5. The origin of preparations for RNA and chip sequencing was somewhat unclear. It would be useful to know if the same cell preparation was split or if preparations came from different animals. More detail is needed to be able to fully assess the results.

6. The manuscript lacks experimental validation of the gene expression changes identified by RNA sequencing, which is crucial. Quantitative PCR, digital droplet PCR or similar methods would enable validation.

7. PCA plots, or similar, are needed to assess the variability among samples of the same genotypes, and relative to control groups. This is crucial data that is lacking from the manuscript.

8. It is difficult to interpret the heatmaps where the data in this study is compared to previously generated data on gene expression. The heatmaps are suggestive of variation between replicates. Venn diagrams would offer a more quantitative assessment of the amount of overlap between the sample sets for this purpose and make it possible to denote the number of genes that show differential expression for both data sets.

9. The potential influence of subpopulations within the tomato-sorted cell population is another concern as olfactory epithelial stem cells also give rise to non-neuronal cell types. RNA from these cell types would also be included in the RNA sequencing. The relative abundance of the different cell types need to be quantified in order to be able to suggest that the effects from lamin B1 knockdown are specific to only a subset of genes involved in the differentiation of a specific cell type. Single-cell RNA sequencing would be a more appropriate choice of method, to be able to draw this type of conclusions.

Lamin B1 is required for mature neuron-specific gene expression during olfactory sensory neuron differentiation

Crystal M. Gigante, Michele Dibattista, Frederick N. Dong, Xiaobin Zheng, Sibiao Yue, Stephen G. Young, Johannes Reisert, Yixian Zheng*, Haiqing Zhao*

Response to Reviewers

We would like to thank the reviewers for their time, feedback, and constructive suggestions. We have addressed the reviewers' concerns point by point below. Changes have been documented in the manuscript in blue text. Aside from changes in response to comments from the reviewers, we have also added a few recent citations to the manuscript.

Reviewer #1:

This is an interesting manuscript seeking to characterize the role of lamin b1 in the development of olfactory neurons.

First, it would be very insightful to delete laminb1 with a mature driver, like OMP-Cre and ask if laminb1 is required for the maintenance of the mature olfactory neuron state.

The reviewer raised an important point. We have attempted to use OMP-Cre (Li et al 2004) to knockout lamin B1 in mature olfactory sensory neurons. We observed clear evidence of recombination of genomic DNA in the olfactory epithelium of the OMP-Cre-driven conditional Lmnb1 mutants (OMP^{Cre/+};Lmnb1^{fl/fl}), but saw no changes in lamin B1 immunofluorescent staining in the olfactory epithelium of the mutant compared to control littermates (OMP^{+/+};Lmnb1^{fl/fl}). We further found no difference in the number of OMP-positive mature neurons, tissue thickness, cell death, or proliferation in OMP-Cre-driven conditional Lmnb1 mutants. A figure summarizing these data can be found at the end of this Response-to-Reviewers. The experiment suggests that lamin B1 protein is not depleted from mature olfactory neurons in OMP-Cre-driven conditional Lmnb1 mutants. In fact, we were unable to observe any evidence of lamin B1 protein depletion in the olfactory epithelium of OMP-Cre-driven knockout mice, even after six months of age. We also attempted to use the Tg#123-CRE (Takeuchi et al 2010) line to drive knockout of lamin B1 in the post-mitotic olfactory lineage, but were similarly unable to see a loss of lamin B1 protein (not shown).

Although we did observe evidence of recombination of genomic DNA, the lack of depletion of Lamin B1 protein in OMP^{Cre/+};Lmnb1^{fl/fl} olfactory epithelium could arise from non-efficient recombination in mature olfactory neurons. It could also lie in the stability of lamins in

postmitotic neurons, which is supported by several lines of evidence in the literature. First, a peptide-labeling screen identified lamin B1 as one of the most stable proteins in neurons of the adult brain (Toyama et al 2013). Another group has identified lamin B1 as a very stable protein in the retina (Razafsky et al 2016). Second, to date, all strategies to knock out *Lmnb1* or *Lmnb2* *in vivo* in mice have been targeted to stem cells or other proliferative cell populations. Employing this strategy, B-type lamin knockout has been successful in the whole embryo (Coffinier et al 2010, Kim et al 2011, Vergnes et al 2004), keratinocytes of the skin (Yang et al 2011), progenitors of the forebrain (Coffinier et al 2011), and stem cells in the olfactory epithelium (this study). No study has yet been published targeting knockout of B-type lamins in postmitotic neurons, which may suggest that lamin B1 protein turnover may be very low in mature olfactory neurons.

We hesitate to include these data in the manuscript because at the moment we are unsure whether the failure to deplete lamin B1 in mature olfactory neurons was caused by lamin stability, incomplete recombination, or both. Regardless the reason, lamin B1 protein is not depleted in the OMP^{Cre/+};Lmnb1^{fl/fl} line, making it obsolete to specifically address the role of Lmnb1 in the maintenance of mature olfactory neurons.

Second, the authors show a reduction of olfactory receptors M72 and P2 expression at later stages of HBC differentiation. I think this results points out to unstable expression of these two genes, which would be the consequence of Adcy3 downregulation (previous reports have shown that Adcy3 is required for stable OR expression). Thus, the authors should also attempt to delete laminb1 also with an ORiresCre line and measure the stability of OR expression in the absence of laminb1. I understand that the second experiment will take time to complete and analyze, especially if the authors do not have these mice at hand. However, it is imperative to at least entertain this possibility in the discussion of the paper.

We agree with the reviewer that knocking out lamin B1 using an ORiresCre would be an intriguing experiment to dissect the role of Lamin B1 in odorant receptor expression. Unfortunately, we do not have access to these mice, and crossing the ORiresCre, Lmnb1^{fl}, and reporter allele into a mouse would take several months and then several additional months for regeneration experiments and analyses. We are even more hesitant to embark on such a task after observing lamin B1 protein persistence in OMP-Cre-driven knockout mice. For these reasons, we would prefer to leave this experiment to a future study. We do, however, consider this point an important one, and have added an expanded discussion of odorant receptor expression in the discussion section of the paper. As we have not performed the necessary experiments to conclude the reason behind the defective odorant receptor expression, we have entertained possible explanations, including the decrease in AC3.

Finally, a minor point is that a lot of the genes that the authors find upregulated in their KO cells are specifically expressed in the Mash1 GBC population (our own unpublished data), show they should tone down the claim that these are genes that are not expressed normally in the olfactory

epithelium and entertain the possibility that these are genes that are only transiently expressed during neuronal differentiation.

The reviewer's point is well taken. We have revised the text to reflect this possibility.

Reviewer #2

The nuclear lamina is a network of proteins that regulates nuclear structure and function and consists of a two-dimensional matrix of mainly A- and B-type lamin proteins located beneath the inner nuclear membrane...

However, I have several concerns, related to technical and/or interpretational problems, with the study:

1. Cre is driven by K5, which is active during embryogenesis and in postnatal tissue. This system would remove lamin B1 from olfactory epithelial stem cells and differentiated neuronal cells during normal neurogenesis and acute regeneration. This removal should be indicated in the cartoon in Fig.1A for the steady-state, with a few of the neurons colored red. Depending on the age of the mice being analyzed, normal neurogenesis occurs in at least 2 waves, one during the first postnatal week and the second at 4 months. Is it possible these waves would be evident as reduced lamin B1 in the western for the steady-state sample group compared to control mice with intact lamin B1 alleles? What is the support for intact lamin B1 alleles during embryonic development of the olfactory epithelia in their system?

We have revised the Fig.1A according to the reviewer's suggestion. We wanted to clarify two points here. First, previous developmental characterization has shown that K5 positive cells are not present in the olfactory epithelium until very late embryonic/early postnatal development ((Holbrook et al 1995, Packard et al 2011). We have observed similar expression patterns in mice carrying the K5-Cre used in this study and a reporter allele, but do not include this data as it has already been published. Second, most studies in the olfactory epithelium have found that K5-expressing stem cells are largely quiescent, except under tissue injury ((Carter et al 2004, Mackay-Sim & Kittel 1991, Schwartz Levey et al 1991, Suzuki & Takeda 1991). Normal neurogenesis and even recovery from some forms of injury, instead, involve globose basal cells, a different pool of stem and progenitor cells that do not express K5. In line with this, we have examined the olfactory epithelium of mice that have not undergone methimazole-induced regeneration and have observed, like others (Iwai et al 2008, Leung et al 2007), that K5 positive stem cells give rise to a very small percentage of the olfactory epithelium during embryonic and early postnatal development. Moreover, we did not observe changes lamin B1 protein levels in undamaged mutant animals by Western blot, despite the likely presence of a minority of Lamin B1 null cells in the epithelium. (Figure 4D).

2. The tomato, lamin B1, and cre transgenes are from different constructs and strain backgrounds, which may result in variation in their sensitivity to the cre recombinase.

Quantification of the overlap between tomato red positive cells and lamin B1 negative cells is necessary. This could be done by counting cells in sections that have been stained for lamin B1 and tomato, as seen in Fig. 1B. In the manuscript, the authors refer to tomato and lamin B1/- cells interchangeably (for example in Fig.1D, 2A, 2B, 2C). Even with the inclusion of such quantification, this is misleading and should be avoided.

We thank the reviewer for this point and acknowledge the concern for differences in recombination efficiency. To reduce the potential influence of strain background, all K5Cre and Lmnbl^{fl} mice were backcrossed into the B6 background for at least 5 generations before combining mice. This information has been added to the methods section. We have quantified the overlap between tomato positive cells and lamin B1 negative cells, and have included the result in Supplementary Figure 1D. Quantification of lamin B1 antibody staining revealed that 93.7% of all Tomato-positive cells were lamin B1 negative, compared to 3.5% of control cells. We have revised the text accordingly, and also added a sentence, “Given the small minority (6.3%) of Tomato-positive cells expressing lamin B1, tomato-positive cells in mosaic mutant epithelium will henceforth be referred to as Lmnbl^{-/-}”, in order to avoid potential misleading.

3. In the odor stimulation experiments, and also in other experiments, the authors used the LaminB1 +/- as the control sample group for the experiments. The inclusion of a control with both Lamin B1 alleles intact would allow for the assessment of effects from haploinsufficiency and establish the normal baseline for the different experiments.

Lmnbl^{+/-} cells were used as a control for certain experiments for several reasons. Within the regenerated olfactory epithelium, cells may be derived from K5-expressing horizontal basal cells, from globose basal cells, or from uninjured tissue (non-regenerated). The only way to ensure that cells were regenerated and derived from horizontal basal cells was to use a K5-Cre and a Cre-driven reporter allele. Given the presence of three alleles in our mutant mice, our mating scheme did not produce mice with K5Cre;Lmnbl^{+/+}. A mating scheme including this genotype of mouse would have greatly decreased the chance of having a sex-matched littermate control animal for each mutant animal.

We did examine K5Cre;Lmnbl^{+/+} littermate mice that lacked the Cre-dependent reporter allele in Western blotting, antibody tissue staining, and whole mount odorant receptor expression analysis. In Western blotting experiments, we did not observe any differences in protein levels between K5Cre;Lmnbl^{fl/+} mice and Lmnbl^{fl/fl} or Lmnbl^{+/+} mice after regeneration. We also observed similar patterns of lamin B1 antibody staining in K5Cre;Lmnbl^{fl/+} mice and Lmnbl^{fl/fl} mice in the olfactory epithelium. Furthermore, Tomato-negative cells in K5Cre;Lmnbl^{fl/fl} mice (presumptive Lmnbl^{fl/fl}) show similar lamin B1 staining as all cells in K5Cre;Lmnbl^{fl/+} mice (presumptive Lmnbl^{fl/+} and Lmnbl^{+/-} cells).

For the odor stimulation experiments, the response of Lmnbl^{+/-} cells is the same as we have seen in Lmnbl^{+/+} olfactory neurons. In one of our current studies (manuscript to be submitted to The Journal of Neuroscience), the average peak receptor current generated by 1 second of 100 μ m each of Cineole and Acetophenone, the same stimulation condition that was

tested in this manuscript, was 108.2 ± 14.8 pA (n=31 neurons) in *Lmnb1*^{+/+} olfactory neurons. The *Lmnb1*^{+/-} olfactory neurons in this study responded with 108.8 ± 11.7 pA (n=35 neurons).

As for the RNAseq and ChIPseq experiments, we do not claim that the gene expression and chromatin state of *Lmnb1*^{+/-} cells would be identical to those of *Lmnb1*^{+/+} cells. Still, we believe that the observed changes in gene expression between the heterozygote cells and homozygous null cells point to a clear role for lamin B1 in the control of these genes.

4. Further information regarding samples processed for RNA and chip sequencing would facilitate the interpretation of results. Specifically, number of samples as well as gender and age.

This information can be accessed by the GEO accessions but has now also been added to the text.

5. The origin of preparations for RNA and chip sequencing was somewhat unclear. It would be useful to know if the same cell preparation was split or if preparations came from different animals. More detail is needed to be able to fully assess the results.

This information can be accessed by the GEO accessions but has now also been added to the text.

6. The manuscript lacks experimental validation of the gene expression changes identified by RNA sequencing, which is crucial. Quantitative PCR, digital droplet PCR or similar methods would enable validation.

We have performed qPCR to examine the expression of 23 genes plus 3 reference genes. Despite the fact that qPCR was performed on mosaic mutant tissue, we were still able to detect changes in gene expression that correlated with the RNAseq data from sorted mutant cells, including decreased expression of all mature neuron genes examined. This data is now included in Supplementary Figure S6A.

7. PCA plots, or similar, are needed to assess the variability among samples of the same genotypes, and relative to control groups. This is crucial data that is lacking from the manuscript.

We have now included this data in Supplementary Figure 5.

8. It is difficult to interpret the heatmaps where the data in this study is compared to previously generated data on gene expression. The heatmaps are suggestive of variation between replicates. Venn diagrams would offer a more quantitative assessment of the amount of overlap between the sample sets for this purpose and make it possible to denote the number of genes that show differential expression for both data sets.

We have replaced most of the heatmaps with Venn diagrams, as suggested (Supplementary Figure 6). We have chosen to leave one of the heatmaps in Supplementary Figure 6 to highlight the variation between replicates.

9. The potential influence of subpopulations within the tomato-sorted cell population is another concern as olfactory epithelial stem cells also give rise to non-neuronal cell types. RNA from these cell types would also be included in the RNA sequencing. The relative abundance of the different cell types need to be quantified in order to be able to suggest that the effects from lamin B1 knockdown are specific to only a subset of genes involved in the differentiation of a specific cell type. Single-cell RNA sequencing would be a more appropriate choice of method, to be able to draw this type of conclusions.

We have included quantification of the different cell types that are tomato positive in K5Cre;Lmnb1^{fl/fl} mice and K5Cre;Lmnb1^{fl/+} controls (the controls used in the experiments in question) (Supplementary Figure 1F). We found no difference in the proportion of different cell types between the two cell populations. Neurons are the major cell population in both tissues, as expected. Indeed, as the reviewer pointed, sequencing from a more homogenous population of cells or single-cell sequencing would be more ideal to reveal the changes in gene expression. We, at the moment, are unfortunately unable to conduct reliable single-cell RNA sequencing. However, within our current RNA-seq data, we were able to detect expression of genes in other major cell populations of the olfactory epithelium whose expression levels did not differ between mutant and control cells. We also found differences in protein expression only for candidate mature neuron markers and not markers of other cell types by Western blotting and immunohistochemistry experiments (Figures 3 & 4, Supplementary Figures 2-4). We believe that these combined data provide adequate evidence that lamin B1 affects neuronal gene expression.

- Carter LA, MacDonald JL, Roskams AJ. 2004. Olfactory horizontal basal cells demonstrate a conserved multipotent progenitor phenotype. *The Journal of neuroscience : the official journal of the Society for Neuroscience* 24: 5670-83
- Coffinier C, Chang SY, Nobumori C, Tu Y, Farber EA, et al. 2010. Abnormal development of the cerebral cortex and cerebellum in the setting of lamin B2 deficiency. *Proceedings of the National Academy of Sciences of the United States of America* 107: 5076-81
- Coffinier C, Jung HJ, Nobumori C, Chang S, Tu Y, et al. 2011. Deficiencies in lamin B1 and lamin B2 cause neurodevelopmental defects and distinct nuclear shape abnormalities in neurons. *Molecular biology of the cell* 22: 4683-93
- Holbrook EH, Szumowski KE, Schwob JE. 1995. An immunochemical, ultrastructural, and developmental characterization of the horizontal basal cells of rat olfactory epithelium. *The Journal of comparative neurology* 363: 129-46
- Iwai N, Zhou Z, Roop DR, Behringer RR. 2008. Horizontal basal cells are multipotent progenitors in normal and injured adult olfactory epithelium. *Stem cells* 26: 1298-306
- Kim Y, Sharov AA, McDole K, Cheng M, Hao H, et al. 2011. Mouse B-type lamins are required for proper organogenesis but not by embryonic stem cells. *Science* 334: 1706-10
- Leung CT, Coulombe PA, Reed RR. 2007. Contribution of olfactory neural stem cells to tissue maintenance and regeneration. *Nature neuroscience* 10: 720-6
- Li J, Ishii T, Feinstein P, Mombaerts P. 2004. Odorant receptor gene choice is reset by nuclear transfer from mouse olfactory sensory neurons. *Nature* 428: 393-9
- Mackay-Sim A, Kittel P. 1991. Cell dynamics in the adult mouse olfactory epithelium: a quantitative autoradiographic study. *The Journal of neuroscience : the official journal of the Society for Neuroscience* 11: 979-84
- Packard A, Schnittke N, Romano RA, Sinha S, Schwob JE. 2011. DeltaNp63 regulates stem cell dynamics in the mammalian olfactory epithelium. *The Journal of neuroscience : the official journal of the Society for Neuroscience* 31: 8748-59
- Razafsky D, Ward C, Potter C, Zhu W, Xue Y, et al. 2016. Lamin B1 and lamin B2 are long-lived proteins with distinct functions in retinal development. *Molecular biology of the cell*
- Schwartz Levey M, Chikaraishi DM, Kauer JS. 1991. Characterization of potential precursor populations in the mouse olfactory epithelium using immunocytochemistry and autoradiography. *The Journal of neuroscience : the official journal of the Society for Neuroscience* 11: 3556-64
- Suzuki Y, Takeda M. 1991. Basal cells in the mouse olfactory epithelium after axotomy: immunohistochemical and electron-microscopic studies. *Cell and tissue research* 266: 239-45
- Takeuchi H, Inokuchi K, Aoki M, Suto F, Tsuboi A, et al. 2010. Sequential arrival and graded secretion of Sema3F by olfactory neuron axons specify map topography at the bulb. *Cell* 141: 1056-67
- Toyama BH, Savas JN, Park SK, Harris MS, Ingolia NT, et al. 2013. Identification of long-lived proteins reveals exceptional stability of essential cellular structures. *Cell* 154: 971-82

Vergnes L, Peterfy M, Bergo MO, Young SG, Reue K. 2004. Lamin B1 is required for mouse development and nuclear integrity. *Proceedings of the National Academy of Sciences of the United States of America* 101: 10428-33

Yang SH, Chang SY, Yin L, Tu Y, Hu Y, et al. 2011. An absence of both lamin B1 and lamin B2 in keratinocytes has no effect on cell proliferation or the development of skin and hair. *Human molecular genetics* 20: 3537-44

Figure. Conditional knockout of Lamin B1 in mature olfactory sensory neurons
 A and B. Antibody staining for Lamin B1 (A) and the mature olfactory neuron marker OMP (B) in the olfactory epithelium from the conditional mutant ($Omp^{Cre/+};Lmnb1^{fl/fl}$) and the control littermate ($Omp^{+/+};Lmnb1^{fl/fl}$). C. PCR of genomic DNA isolated from olfactory mucosa (Olfactory), brain, or tail samples from lamin B1 conditional mutant and control littermate. Mouse genotype is given in rows marked OMP and Lmnb1. Amplification of Lmnb1^{fl} allele, Lmnb1 knockout allele (recombined Lmnb1fl allele, Lmnb1⁻), and wildtype allele (Lmnb1⁺) are shown by colored asterisk. Bands below 500 bp are non-specific. Amplification of the knockout allele was only ever observed in mutant samples from the olfactory epithelium. D. Quantification of cell proliferation in the olfactory epithelium of the conditional mutant and the control littermate 30 days after birth (P30). Proliferation rate determined by EdU retention after a 24 hour pulse. E. Number of cells per millimeter of olfactory epithelium in the conditional mutant and the control at postnatal day 10 (P10). F. Cell death, measured by antibody staining for cleaved caspase 3 (CC3), in the conditional mutant and the control. G. Tissue thickness of the olfactory epithelium of the OMP-driven Lmnb1 conditional mutant and littermate control at P10 and P30.

REVIEWERS' COMMENTS:

Reviewer #1 (Remarks to the Author):

The authors took the critiques seriously and made every effort possible to respond. It is unfortunate that deletion by OMPiCre fails to deplete lamin B1 from mature olfactory neurons but it is great to include these data in the manuscript because the stability of lamin B1 in the postmitotic neurons is an interesting finding. The manuscript adds significant insight to the role of the nuclear lamina in the differentiation of olfactory neurons, especially in light of recent observations (LeGros et al., 2016) showing that there is some heterochromatin left at the nuclear lamina of OSNs. Thus lamin B1 may play a complementary role to lamin B receptor. I support publication of the current form of this manuscript.

Reviewer #2 (Remarks to the Author):

The authors have satisfactorily responded to my questions and made the necessary changes to the manuscript.

Lamin B1 is required for mature neuron-specific gene expression during olfactory sensory neuron differentiation

Crystal M. Gigante, Michele Dibattista, Frederick N. Dong, Xiaobin Zheng, Sibiao Yue, Stephen G. Young, Johannes Reisert, Yixian Zheng*, Haiqing Zhao*

Response to Reviewers

Reviewer #1:

... .. It is unfortunate that deletion by OMPiCre fails to deplete lamin B1 from mature olfactory neurons but it is great to include these data in the manuscript because the stability of lamin B1 in the postmitotic neurons is an interesting finding.

We thank the reviewer for his/her time and suggestions. We have included the *Omp^{Cre/+};Lmnb1^{fl/fl}* results in the supplementary information as per the reviewer's suggestion.

Reviewer #2

The authors have satisfactorily responded to my questions and made the necessary changes to the manuscript.

We thank the reviewer for his/her time in reviewing the revised manuscript.